# Indirect Fortification of Traditional Nixtamalized Tortillas with Nixtamalized Corn Flours

**DOI:** 10.3390/foods13244082

**Published:** 2024-12-17

**Authors:** María Guadalupe Nieves-Hernandez, Brenda Lizbeth Correa-Piña, Oscar Garcia-Chavero, Salomon Lopez-Ramirez, Rosendo Florez-Mejia, Oscar Yael Barrón-García, Elsa Gutierrez-Cortez, Marcela Gaytán-Martínez, Juana Isela Rojas-Molina, Mario E. Rodriguez-Garcia

**Affiliations:** 1Research and Graduate Program in Food Science, School of Chemistry, Universidad Autónoma de Querétaro, Cerro de las Campanas S/N, Col. Centro, Santiago de Querétaro C.P. 76010, Querétaro, Mexico; mghernandeez@gmail.com (M.G.N.-H.); correapilizbeth@gmail.com (B.L.C.-P.); marcelagaytanm@yahoo.com.mx (M.G.-M.); 2Universidad Tecnológica del Norte de Guanajuato, Carretera Victoria-San Luis de la Paz km 1 Col. Paso Hondo, Victoria C.P. 37920, Guanajuato, Mexico; oscar.garcia@utng.edu.mx (O.G.-C.); salomon.lopez@utng.edu.mx (S.L.-R.); rosendo.flores@utng.edu.mx (R.F.-M.); 3Departamento de Nanotecnología, Centro de Física Aplicada y Tecnología Avanzada, Universidad Nacional Autónoma de México, Campus Juriquilla, Santiago de Querétaro C.P. 76230, Querétaro, Mexico; 4División Industrial, Universidad Tecnológica de Querétaro, Av. Pie de la Cuesta 2501, Nacional, Santiago de Querétaro C.P. 76148, Querétaro, Mexico; 5Laboratorio de Procesos en Ingeniería Agroalimentaria, Unidad de Investigación Multidisciplinaria (UIM), Universidad Nacional Autónoma de México, FES-Cuautitlán, Cuautitlán Izcalli C.P. 54714, Estado de México, Mexico; elsaneqpm@yahoo.com.mx; 6Laboratorio de Química Medicinal, Facultad de Química, Universidad Autónoma de Querétaro, Cerro de las Campanas S/N, Santiago de Querétaro C.P. 76017, Querétaro, Mexico; jirojasmolina@gmail.com

**Keywords:** tortilla, fortification, minerals, vitamins, nixtamal

## Abstract

Background: This work focused on the study of the indirect fortification of Mexican tortillas made from nixtamalized masa (NM) with nixtamalized commercial corn flour (NCC-F) fortified with Zn, Fe, vitamins and folic acid. Methods: The chemical proximate values (CPV), ash content, mineral composition by inductively coupled plasma, in vitro protein digestibility (PD), protein digestibility-corrected amino acid score (PDCAAS), the total starch content, the resistant starch (RS) content in nixtamalized corn tortillas (NC-T) and nixtamalized commercial corn flour tortillas (NCCF-T) and the contribution of tortillas prepared with a mixture of NM and NCC-F (75:25 and 50:50, NM:NCC-F) to the recommended dietary intake (RDI) of minerals and vitamins were determined. Results: No significant differences (*p* < 0.05) were found in CPV and RS content between NCCF-T and NC-T. Ca content was significantly higher (*p* < 0.05) in NC-T than in NCCF-T, while Fe, K, Zn, folic acid contents, PD and PDCAAS content was higher in NCCF-T compared to NC-T (*p* < 0.05). The tortillas made with a mixture of NM and NCC-F (50:50) provide 43.07% of the RDI of Ca for Mexican children and adults, while ~45% and >100% of the RDI of Mg for adults and children, respectively, are provided by these tortillas. Similarly, tortillas from the NM:NCC-F mixture (50:50) provide average values of 45, 71 and ~91% of the RDI of Fe, Zn and folic acid, respectively, for all age groups of the Mexican population. Conclusions: NCCF-T contribute significantly to the recommended daily intake (RDI) of micronutrients such as iron, zinc, magnesium and folic acid, while tortillas made from the traditional nixtamalized corn of the Mexican diet have a higher calcium content. Tortillas made from a mixture of NCC-F and traditional NM may be an effective way to address micronutrient deficiencies in the Mexican population.

## 1. Introduction

The corn tortilla is a staple food in the diet of the Mexican population and represents an element of cultural and gastronomic identity [1]. It is estimated that the per capita consumption of tortillas in Mexico among the urban population is around 85 kg per year; however, this consumption may be higher among the rural population. In this context, it is reported that about 11.6 million tons of tortillas were produced in the country in 2020 [2]. These data reflect a high and sustained demand for tortillas, which are satisfied by traditional nixtamalization and the use of nixtamalized corn flour, which is why this product is considered an element of food security in our country [3].

Nixtamalization is a traditional process that has been used for centuries, especially in Mesoamerica, to prepare maize for consumption. In this process, the corn kernels are boiled and soaked in an alkaline solution, usually water with lime (calcium hydroxide), and then rinsed and ground. The alkaline treatment improves the nutritional profile of maize by increasing the availability of niacin (vitamin B3), improving protein quality and reducing toxins such as mycotoxins. It also gives the maize a distinct flavor and texture, making it suitable for various culinary applications. The term “masa” refers to the dough made from nixtamalized maize. After the kernels are ground, they form a pliable dough that serves as the base for a variety of traditional foods, including tortillas, tamales and pupusas. Masa plays a central role in the diet and culture of many regions, especially in Latin America, where it is a staple food [4].

The nixtamalized masa (NM) from the traditional method accounts for two-thirds of tortilla production. In thousands of tortillerias, nixtamalized masa is mixed with nixtamalized corn flour, which in most cases improves the malleability and yield of the masa [5]. The Mexican government has regulated the addition of supplements and additives in these staple foods to protect the health of consumers [6,7] and through the proposed non-official [8].

Fresh masa and dry masa flour are becoming increasingly popular worldwide, as this process is used to make tortillas and fried snacks such as corn chips and tortilla chips, which are consumed all over the world. Three important products are made industrially from nixtamalized corn: tortillas, corn chips, and tortilla chips [9,10].

In Mexico, consumers are not aware of the contribution of nixtamalized corn flour in indirect fortification to the recommended daily intake (RDI) of vitamins and minerals for the Mexican population. The fortification of flours with vitamins (folate) and minerals (Ca, Zn and Fe) can be an effective way to combat micronutrient deficiencies [11]. However, this requires specific amounts and types of supplements to ensure that the diet contains an adequate amount of nutrients.

The designation of corn flour provided by the official Mexican standard [8] prescribes the percentages of folic acid, zinc, iron, vitamins B1 and B2 and niacin with which nixtamalized corn and nixtamalized corn flour must be fortified; however, this regulation does not allow the consumer to know whether the corn tortilla is sufficiently fortified to be nutritious. The commercially available nixtamalized corn flours for tortillas are fortified, while the nixtamalized tortillas made from fresh masa are not fortified [11]. This can be confusing for consumers, as tortillas made with 100% corn are often not packaged and labeled, making it difficult to distinguish between the quality of the masa and the flour used. On the other hand, tortillas made with other ingredients besides corn are packaged and differentiated by the information on the label that the standard requires [12].

In Mexico, the concept of tortilla quality is still debated, as it is known that consumer preferences depend mainly on the tastes and habits of the geographical regions [13]. In northern Mexico, the white tortilla is considered the better one because it is lighter and softer, as it contains little calcium and is usually made from nixtamalized commercial corn flour. In central Mexico, on the other hand, most tortillas are made in the traditional way, which means that the tortilla has a yellowish appearance due to its high calcium content [14]. In most cases, tortillas in both regions are made from a mixture of traditional and nixtamalized corn flour [15].

Shamah-Levy et al. [16] described that the prevalence of iron and vitamin B12 deficiency in women of reproductive age (20–49 years) is still a problem in Mexico, while folic acid deficiency is no longer a problem. Iron and zinc deficiencies require prevention and fortification strategies. Therefore, the indirect fortification of traditional tortillas with nixtamalized corn flour could help to reduce this problem.

Morales and García [17] studied the nutritional composition of tortillas made using the traditional method and nixtamalized commercial corn flour and they found differences in carbohydrate and iron contents attributed to the phenotype of the corn. The main differences between the two methods are that in the traditional nixtamalization method, the soaking time is about 0 to 24 h, and the lime concentration is 2%, while in the commercial nixtamalization method, the soaking time is about 1 to 3 h, and the lime concentration is 0.5 to 1%. Therefore, the commercial method is faster and uses 50% less lime concentration. Likewise, these authors reported that the corn flour tortilla (Masa Flour—MFA) contained fewer nutrients than the traditional masa tortilla (Masa Dough—MDB); however, the levels of crude fat, fiber, thiamine and riboflavin were twice as high or higher in MFA, which can be explained to the supplements and additives added by the manufacturers.

According to Orjuela et al. [18], there are strategies in Mexico to improve folic acid and folate intake in foods based on baked goods and corn porridge. Fortification was introduced in 2001, but without direct enforcement. The authors pointed out that uncontrolled fortification of staple foods with folic acid leads to unpredictable total folate intake without adequately influencing the intended target, which is why mandatory fortification is needed.

Since 2016, the FDA has allowed manufacturers to voluntarily add up to 1.54 mg/kg of folic acid to cornmeal. Currently, nineteen countries in Latin America add folic acid in concentrations between 1.5 and 3.4 mg/kg in flour. Folic acid, a synthetic form of folate, is a B vitamin whose consumption in pregnant women can help to prevent neural tube defects, i.e., congenital malformations affecting the brain, spine and spinal cord [19].

On the other hand, a high prevalence of micronutrient deficiencies has been found in all age groups of the Mexican population. For example, schoolchildren consume less calcium, vitamin A and vitamin B1 (thiamine) than recommended. In adolescents; calcium, magnesium; vitamins A, B1, B2, B12 and C; and folic acid are the nutrients with the highest prevalence of inadequate intake, while adults have the highest prevalence of inadequate intake for most nutrients, and vitamin D deficiency is 100% in all age groups [20]. Mexico is one of the top five countries with the highest consumption of maize in the world [21], which is mainly consumed in the form of tortillas. This fact represents a strategy to overcome the micronutrient deficiency in the population [22]. Wheat and maize flour fortification is a preventive strategy aimed at improving micronutrient levels in the population. It is integrated into the framework of other interventions to reduce vitamin and mineral deficiencies in the population when these are recognized as a public health problem [23].

In Mexico, there are various indirect strategies for enriching tortilla and nixtamalized corn flour. Cornejo-Villegas et al. [24] fortified nixtamalized corn flour with nopal (*Opuntia ficus indica*) flour between 0 and 10%. The addition of 4% nopal powder increased the calcium and fiber content of nixtamalized maize flour by 300 and 26.31%, respectively. This fact is very important from a nutritional point of view, as tortillas are the main source of minerals and fiber in some areas of Mexico and meet the daily requirements for Ca intake. In addition, Contreras et al. [25] added chapulin (*Sphenarium magnum*) flour to nixtamalized corn flour at concentrations of 2, 6 and 10% (*w*/*w*), and the results showed a significant increase in protein and mineral content. When the chapulin flour in the tortilla recipe is increased from 2 to 10%, the S content increases from 6.56 mg/100 g to 32.81 g/100 g and the Zn content from 0.228 mg/100 g to 1.14 mg/100 g. The main benefits of adding chapulin flour were the increase in protein content, essential mineral content and chitin content. However, the addition of chapulin flour to cornmeal in concentrations of more than 6% has significant effects on the viscosity and hardness of the corn masa.

Against this background, we hypothesized that the tortillas consumed in Mexico have better nutritional quality thanks to the indirect fortification of the NM with NCC-F.

The aim of this work was to investigate the contribution of the addition of nixtamalized corn flour to the micronutrient content of corn tortillas, which is a staple food consumed by the Mexican population. The ash content, macro and mineral nutrients, riboflavin, thiamine, folic acid, protein digestibility, PDCAAS and resistant starch in tortillas made with nixtamalized masa and nixtamalized commercial corn flour were investigated.

## 2. Materials and Methods

### 2.1. Sample Obtention

The tortillas were sourced from 18 different tortilla factories. The first collection of tortillas was obtained in March 2023 and the second collection in April 2023. The locations of the tortilla factories studied are listed in Table 1. The nixtamalized commercial corn flour (NCC-F) purchased from a supermarket in Queretaro City in March 2023 was used to make the tortillas. The theoretical nutritional composition of corn flour, as stated by the manufacturer on the product label, is shown in Table 2. As can be seen, the nixtamalized corn flour is enriched with vitamins and minerals.

### 2.2. Chemical Proximate Analysis

The protein, fat, moisture, ash and total dietary fiber contents of the samples were analyzed following AOAC (2000) official methods 979.06, 920.39, 925.10, 923.03 and 992.16, respectively [26].

### 2.3. Mineral Composition by Inductively Coupled Plasma-Optical Emission Spectrometry (ICP-OES)

The final mineral content was determined according to the method proposed by Ramirez-Gutierrez et al. [27]. The tortillas were analyzed using an inductively coupled plasma optical emission spectrometer (ICP-OES) (model Thermo iCAP 6500 Duo View, Thermo Fisher Scientific, Waltham, MA USA). An amount of 5 g of the sample was converted to white ash by incineration in a furnace at 550 °C for 5 h. Subsequently, 0.1 g of the ash was digested with nitric acid (Baker 69–70%); this step was performed in triplicate. The digested samples were filtered and then exposed to an argon plasma to excite the elements identified by their characteristic emission spectra. The emission intensities were later converted into elemental concentrations by comparing them with the standard curves.

### 2.4. Vitamin Content

The vitamin content was measured using a liquid chromatographic method for the simultaneous determination of thiamine, niacin, riboflavin and folic acid in NC-T and NCCF-T. Ion-pair chromatography with a C_18_ reversed-phase column was used. Four vitamins were separated in a single analysis; the total analysis time never exceeded 55 min. A mobile phase of methanol–water (15:85) and 5mM octanesulfonic acid with 0.5% triethylamine at a pH of 3.6 and a flow rate of 1.0 mL/min gave the most satisfactory separation of these vitamins using a UV detector tuned to different wavelengths. Sample preparation included acidification to precipitate the proteins and centrifugation followed by gravity filtration. Line fidelity, precision, recovery and sensitivity were always satisfactory. The detection limits were in the range of 0.02 to 0.10 μg/mL and the limits of quantification were in the range of 0.03 to 0.25 μg/mL.

### 2.5. In Vitro Protein Digestibility

The in vitro method described by Hsu et al. [28] was used. A multienzyme system consisting of porcine pancreatic trypsin (Type IX, 15,310 units/mg protein), bovine pancreatic chymotrypsin (Type II, 48 units/g solid), porcine intestinal peptidase (P-7500, 115 units/mg solid) and bacterial protease (Type XIV, 4.4 units/mg solid) (Sigma-Aldrich Co., St. Louis, MO, USA) was used. All samples were sieved through a No. 60 sieve (250 µm). An amount of 50 mL of the aqueous protein suspension (6.25 mg protein/mL) was prepared in distilled water, adjusted to pH 8.0 with HCl and/or NaOH (0.1 N) and stirred in a water bath at 37 °C. The multienzyme solution (1.6 mg trypsin, 3.1 mg chymotrypsin and 1.3 mg peptidase/mL) was kept in an ice bath and adjusted to pH 8.0 with HCl and/or NaOH (0.1 N). Subsequently, 5 mL of the multienzyme solution was added to the protein extract while stirring at 37 °C. The pH value was recorded over a period of 10 min using a potentiometer.

The pH was recorded after 1 min and used to estimate the in vitro protein digestibility (IVPD) according to Equation (1) below:(1)$$IVPD\;\left(\%\right)=229.30-30.30{\;pH}_{20}$$
where *p**H*_20_ is the pH value of the suspension after 20 min of digestion.

### 2.6. Total and Resistant Starch

The resistant starch content in the NCT and NCCF-T was determined using a commercial kit (R-Starch, Megazyme^®^, Bray, Ireland) according to the method described by Rojas-Molina et al. [29]. First, the samples were subjected to protein hydrolysis with pepsin (3200–4500 U/mg, Sigma Chemical Co., St. Louis, MO, USA) in an acidic medium (pH = 2.0) and incubated for 30 min at 37 °C in a shaking water bath. Then, the samples were mixed with pancreatic α-amylase (3 Ceralpha U/mg) for 16 h at a neutral pH. The samples were then centrifuged at 1500× *g* for 10 min, and the hydrolysis products were removed. The residue (indigestible starch fraction) was then dispersed in alkaline medium (2 mL of 2 M KOH for each sample, J.T. Baker Center Valley, PA, USA) and hydrolyzed with amyloglucosidase (300 U/mL). Subsequently, 8 mL of 1.2 M sodium acetate buffer (pH = 3.8, J.T. Baker Center Valley, PA, USA) was added to each sample with stirring. Immediately afterwards, 0.1 mL of amyloglucosidase (300 U/mL) was mixed into the samples. The samples were placed in a water bath at 50 °C and incubated for 30 min with intermittent mixing using a vortex mixer. The samples were then transferred to a 100 mL volumetric flask, made up to 100 mL with distilled water and mixed. An aliquot of 10 mL was then centrifuged at 1500× *g* for 10 min (Heraeus, Thermo Scientific, Bartlesville, OK, USA). Two aliquots of 0.1 mL each were transferred to glass test tubes and 3.0 mL of the reagents glucose oxidase, peroxidase and 4-aminoantipyrine (GOPOD) were added to the test tubes. The tubes were then incubated at 50 °C for 20 min. Finally, the absorbance of each solution was measured at 510 nm (VE-5100 UV, Velab, Pharr, TX, USA) against the reagent blank (0.1 mL of 100 mM sodium acetate buffer at pH = 4.5 and 3.0 mL of GOPOD reagent) and compared to the absorbance of D-glucose standards prepared with 0.1 mL of D-glucose (1 mg/mL) and 3 mL of GOPOD. In addition, the total starch content in the isolated starches was determined using a commercial kit (K-TSTA-100A, Megazyme^®^, Bray, Ireland) according to the instructions of the manufacturer. Starch yield and starch separation efficiency were calculated as follows: Starch yield (%) = {[weight of isolated starch (g) × 100%]/weight of corn sample (g)}. Starch separation efficiency = {[weight of isolated starch (g) × 100%]/total starch in the maize sample (g)} [30].

### 2.7. RDI Contribution in 50/50% NC-T/NCCF-T

Since there are no official data on the amount of nixtamalized corn flour mixed with the traditional nixtamalized masa, a brief exploratory survey was conducted in the main tortilla outlets in Mexico City and the metropolitan area. It was found that most tortilla suppliers use at least a 50:50 mix. These data were then used to calculate the contribution of the tortilla to the RDI of the micronutrients contained in this food. On the other hand, it was found that the average per capita consumption of tortilla in Mexico is around 240 g/wet ground (129.84 g/dry ground) consisting of a 50:50 mix (nixtamalized corn flour:traditional nixtamalized corn).

### 2.8. Statistical Analysis

The mean and standard deviations of three independent experiments are shown. Data were analyzed by analysis of variance (ANOVA) and comparison of means by Tukey’s test (α = 0.05) using MiniTab 2018 statistical software (Minitab Inc., State College, PA, USA).

## 3. Results and Discussion

### 3.1. Proximal Analysis

Table 3 shows that there are no significant statistical differences in the moisture and protein content of the two tortillas (*p* < 0.05). There is a significant statistical difference in ash content because the traditional method requires a long soaking time, which allows for Ca to penetrate the pericarp, germ and endosperm [31], while the short cooking time of the nixtamalized corn flour reduces the Ca content. However, the ash content in the case of NCCF-T contains iron and zinc (Table 2). On the other hand, Ca is intentionally added to the NM so that the masa has a longer shelf life. This means that the ash content in corn tortillas is the result of intrinsic and extrinsic processes. There is no statistical difference in the total dietary fiber content (soluble and insoluble fiber) between the tortillas tested. From a nutritional point of view, the higher soluble fiber content in NCCF-T is important, and these results suggest that the lower content reported for NCCF-T was corrected for the factories. Finally, the fat values showed no significant differences between the tortilla samples analyzed.

### 3.2. Mineral Content

Table 4 shows the macro- and microminerals found in NC-T and NCCF-T. The contribution of minerals (Ca, P, Mg, Fe and Zn) of the components in tortillas made with a mixture of fortified nixtamalized commercial corn flour and nixtamalized masa was calculated using the data in Table 5, while the contribution of vitamins and folic acid was calculated using the data in Table 6 and according to the recommended daily intake for the Mexican population [32]. In the samples analyzed, the Ca content is higher in NC-T than in NCCF-T. As previously mentioned, the long soaking time in the traditional process allows for the diffusion of Ca into the different corn components (pericarp, germ and endosperm), and a high Ca(OH)_2_ content was used in this process—up to 2% in some cases. In contrast, the Ca content of corn meal during cooking is between 0.5 and 1%. The P content does not differ between the samples and corresponds to the endogenous content. The same is true for Mg, K, S and Na, but in NCCF-T, the Fe and Zn content is significantly higher than in NC-T. As already mentioned, this fact corresponds to the enrichment of this sample, and of course, the production of hybrid tortillas from both samples also contributes directly to the enrichment of the final product.

### 3.3. Vitamin Content

Table 6 shows the vitamin content in the NC-T and NCCF-T samples. The results show significant differences (*p <* 0.05) between NC-T and NCCF-T for thiamine, riboflavin, niacin and folic acid. This shows that there are obvious differences between the tortillas tested. A higher content of niacin, riboflavin and thiamine was found in NCCF-T compared to NC-T. On the other hand, the folic acid content was significantly higher in NCCF-T compared to NC-T, which can be attributed to the Mexican standards for the addition of food supplements.

### 3.4. Amino Acid Content, In Vitro Protein Digestibility and PDCAAS

The results of the in vitro protein digestibility of NC-T and NCCF-T nixtamalized tortilla proteins are shown in Table 7. The nutritional quality of a protein is determined by the amino acid composition and its digestibility. In vitro digestibility is an indicator of the nutritional quality of proteins [33]. Maize has a low nutritional protein value due to its limiting amino acids, lysine and tryptophan. However, nixtamalization increases the biological value of the proteins in nixtamalized tortillas [34]. Significant differences were found between the tortillas since the NCCF-T had a 10% higher digestibility than the NC-T. This could be due to the processing conditions, which are most aggressive during the traditional nixtamalization process due to the lime concentration. During nixtamalization and tortilla making, temperatures ranging from 80 to 90 °C during cooking and over 200 °C during baking significantly impact the lysine and tryptophan content, leading to losses of approximately 18.60% and 21.47%, respectively. These losses occur primarily due to the Maillard reaction, where lysine reacts with reducing sugars and the alkaline hydrolysis and thermal degradation of amino acids, particularly tryptophan. The high pH environment and elevated temperatures accelerate these reactions, reducing the nutritional quality of the final product. Mitigating strategies include optimizing process conditions, such as lowering temperatures and cooking times, and fortifying tortillas to compensate for the loss of these essential amino acids [35]. On the other hand, the study shows that the PDCAAS (protein digestibility-corrected amino acid score) index is consistently higher in NCCF-T, indicating a higher quality of protein with respect to the amino acid requirements of infants, children and adults compared to NC-T.

### 3.5. Total and Resistant Starch

The results showed higher values in resistant starch content in NCCF-T compared to NC-T. Rojas-Molina et al. [29] related the resistant starch content to the part of the starch that is not gelatinized during the nixtamalization process (native starch) and is associated with RS_2_, in addition to RS_3_, which is formed by the retrogradation process of the gelatinized starch. Both types contribute to the total content of resistant starch in nixtamalized tortillas.

### 3.6. Contribution of NCCF-T and NC-T to the Recommended Daily Intake (RDI) of Main Micronutrients According to Experimental Data

In relation to tortilla consumption, Table 8 shows the contribution of tortillas made with a mixture of fortified nixtamalized commercial corn flour and nixtamalized masa to the recommended daily intake (RDI) of vitamins and minerals for the Mexican population. This considers that the average per capita consumption of tortillas in Mexico is 240 g/day (the dry matter content is 129.84 g) and that one-third of the tortilla suppliers use a mixture of 75:25 percent by weight of traditionally nixtamalized corn:industrially nixtamalized corn flour [36]. However, this percentage varies up to a mixture of 50:50 percent by weight, which is a decision of the tortilla suppliers. There are no official data, but it is estimated that 70% of tortilla suppliers in Mexico make tortillas mainly from traditionally nixtamalized corn, while less than 30% use mainly industrially nixtamalized corn flour [37].

Table 8 shows that the 50:50 tortilla for children and adults provides 43.07% of the RDI of calcium (344.56 mg/day), while the tortilla for adolescents and pregnant and breastfeeding women provides 34.46 and 28.71% of the RDI, respectively. Regarding phosphorus, tortilla consumption contributes to 54.90, 21.96 and 39.21% of the RDI for children, adolescents and adults (including pregnant and breastfeeding women), respectively. Adequate dietary calcium intake is necessary for the development of peak bone mass in childhood, the maintenance of bone health in adulthood, and the reduction of bone loss in postmenopausal and aged women [38]. In general, milk and dairy products are the main sources of calcium in the Western diet, accounting for 43.4% of total calcium consumption [39]. In Mexico, milk consumption has become less important, mainly due to the tendency to replace it with a variety of non-dairy products, so that per capita milk consumption in Mexico ranks 60th in the world [40]. This also has an impact on the diet, making the tortilla the most important source of calcium for the Mexican population according to the data in Table 8.

On the other hand, the mass ratio of calcium:phosphorus in the tortilla is 1:3. The absorption of the Ca:P ratio is of interest because both minerals interact in the gastrointestinal tract and limit the absorption of the other minerals. For an optimal supply of calcium and phosphorus for bone health, a Ca:P mass ratio of about 1–2:1 is recommended [41]. In the case of tortillas, this ratio is appropriate. Tortillas provide ~45% of the Mg requirements of the adult Mexican population and breastfeeding women and ~50% in the case of pregnant women. For children and adolescents, this staple food provides >100% and 40% of the daily Mg requirement, respectively (see Table 8). The consumption of less than the recommended amount of Mg and Ca is associated with lower bone mineral density and higher fracture risk. Recently, the prevalence of osteopenia and osteoporosis has increased in the economically active Mexican population, even in young people [42]. Therefore, tortilla consumption is an aid in the prevention of these diseases. Table 8 shows that tortillas provide 74.2% of the zinc RDI for the Mexican population (children, adolescents, adults and pregnant women). In the late 1990s, the 1999 Mexican National Health and Nutrition Survey (ENN 99) reported a high prevalence of zinc deficiency (25.3%) in children under 11 years of age. Public interventions with zinc were initiated and dietary supplements with various micronutrients, including zinc, were tested in a clinical trial in urban Mexican preschool children; however, the results showed that the prevalence did not decrease between 1999 and 2006 [43]. The data presented in Table 8 show that tortillas made with zinc-fortified flour continue to be an appropriate strategy to combat this mineral deficiency in the population.

In Mexico, iron deficiency anemia (IDA) has a medium public health significance (20.0–39.9%) according to the World Health Organization (WHO) scale [44]. Women of childbearing age, breastfeeding women and preschool children are the groups most at risk [45,46]. In recent years, public health policies have enabled a slight decrease in the prevalence of IDA, although it has increased again in recent years [47]. Table 8 shows that tortilla consumption provides 50% of the RDI for iron for preschool children, adolescents, adults and breastfeeding women and 25% of the RDI for pregnant women. It is likely that the recent increase in the prevalence of IDA is due to the dietary transition in Mexico, where foods high in iron are being replaced by foods high in energy and deficient in micronutrients [48].

An adequate intake of folic acid before pregnancy and in the first weeks of pregnancy protects newborns from neural tube defects [49]. For this reason, twelve countries worldwide, including Mexico, prescribe the fortification of corn flour and meal with folic acid and other micronutrients [6,50]. Tortilla consumption provides >100 of the RDI of folic acid for children, adults and breastfeeding women, while for adolescents and pregnant women, this food provides 100 and 56% of the RDI, respectively (see Table 8). Fortification of cornmeal with folic acid has been reported to reduce neural tube defects in the Hispanic population in the United States [51]. However, some vulnerable populations in Mexico (rural areas) still have deficiencies of this micronutrient and associated problems [52]. It is likely that the main beneficiaries of fortified flour are those who live in urban areas or have higher socioeconomic advantages, where the consumption of tortillas made with 100% fortified corn flour is more widespread [53].

The results presented in Table 8 show that tortillas consumed in the central zone of Mexico do not provide the RDI of vitamins (B1, B2 and B3) expected by a public health program. It is known that tortillas in Mexico are often made from a mixture of dried, nixtamalized corn and traditional masa nixtamal (wet). This is because the mixture of traditional masa nixtamal with fortified nixtamalized corn flour improves the texture, extends the shelf life, improves the taste and reduces production costs [48].

## 4. Conclusions

Indirect fortification of corn tortillas by mixing nixtamalized masa with fortified nixtamalized corn flour could be a suitable strategy to reduce micronutrient deficiencies in the Mexican population without changing their dietary habits. Taste preferences play a critical role in the success of fortified tortillas as a strategy to address micronutrient deficiencies. If fortified tortillas differ significantly from conventional tortillas in taste, texture or appearance, they may be rejected by consumers, limiting their acceptability and impact. The tortillas made with nixtamalized corn flour contribute significantly to the RDI of micronutrients such as iron, zinc and folic acid, while tortillas made with traditional nixtamalized corn have a higher calcium content in the Mexican diet. However, studies on physicochemical properties and sensory acceptability are essential to ensure that fortification does not alter the properties that comprise tortillas. To increase the calcium content in fortified tortillas, more calcium hydroxide could be used in the nixtamalization process or calcium-rich additives could be incorporated into the fortified flour. These adjustments must maintain taste, texture and overall acceptability while increasing nutritional value and ensuring consumer acceptance. In addition, the regulations that govern the production of tortillas by manufacturers should be reviewed to ensure that the manufacturing practices of this staple food do not conflict with public policy efforts to reduce micronutrient deficiencies in the Mexican population.

## Figures and Tables

**Table 1 foods-13-04082-t001:** Location of tortilla factories (NCCF-T, and NC-T).

Time (Months)	Location	Sampled Establishments	
		NC-T	NCCF-T
1	Querétaro City	3	3
2	Querétaro City	3	3
1	Cuautitlán Itzcalli	3	3
2	Cuautitlán Itzcalli	3	3
1	City of Victoria	3	3
2	City of Victoria	3	3

NCCF-T: Nixtamalized commercial corn flour tortilla, NC-T: Nixtamalized corn tortilla.

**Table 2 foods-13-04082-t002:** Theoretical nutritional information of NCC-F (per 100 g of flour).

**Nutritional Information**		**Content (per 100 g)**
Energy content		1457 kJ (344 kcal)
Protein		8 g
Fat		4 g
Saturated		0 g
Monounsaturated		1 g
Polyunsaturated		2 g
Trans fatty acids		0 g
Cholesterol		0 mg
Carbohydrates		75 g
Sugars		1 g
Dietary fiber		6 g
Sodium		0 mg
**Micronutrients**	**Content (mg/100 g)**	**% VNR ***
Calcium	49.3	5.4
Iron	4	23.5
Zinc	4	40
Vitamin B1 (Thiamin)	0.5	62.5
Vitamin B2 (Riboflavin)	0.3	35.7
Vitamin B3 (Niacin)	3.5	31.8
Folic acid (Folacin)	0.2	52.6

* VNR references nutritional value for Mexican population according to NOM-051-SCFI/SSA1-2010. NCC-F: Nixtamalized commercial corn flour.

**Table 3 foods-13-04082-t003:** Proximal analysis of tortillas elaborated with traditional nixtamalized corn (NC-T) and nixtamalized corn flour (NCCF-T).

Tortilla	NC-T	NCCF-T
Tortilla moisture (%)	43.40 ± 3.03 ^a^	45.63 ± 3.90 ^a^
Moisture (dry sample) (%)	5.79 ± 1.56 ^a^	5.40 ± 1.54 ^a^
Total protein (%)	8.58 ± 0.46 ^a^	8.49 ± 0.20 ^a^
Ash (%)	2.17 ± 0.57 ^a^	1.79 ± 0.33 ^b^
Total dietary fiber (%)	12.67 ± 3.47 ^a^	12.44 ± 3.26 ^a^
Fat (%)	1.73 ± 0.49 ^a^	1.73 ± 0.42 ^a^

The values represent the mean and standard deviation (SD), n = 5. Mean values in rows with different letters differ significantly (*p* ≤ 0.05). NCCF-T: Nixtamalized commercial corn flour tortilla, NC-T: Nixtamalized corn tortilla.

**Table 4 foods-13-04082-t004:** Mineral content in tortillas elaborated with traditional nixtamalized corn (NC-T) and nixtamalized commercial corn flour (NCCF-T).

Sample	NC-T (mg/100 g)	NCCF-T (mg/100 g)
Ca	343.39 ± 76.52 ^a^	82.00 ± 28.71 ^b^
P	230.98 ± 43.68 ^a^	215.88 ± 54.23 ^a^
K	221.46 ± 40.40 ^b^	319.40 ± 24.88 ^a^
Mg	131.83 ± 31.80 ^a^	143.52 ± 55.35 ^a^
S	15.64 ± 8.71 ^a^	5.64 ± 3.20 ^b^
Na	9.72 ± 3.06 ^b^	19.68 ± 5.32 ^a^
Zn	1.95 ± 0.30 ^b^	7.21 ± 1.02 ^a^
Si	3.09 ± 1.84 ^a^	2.02 ± 0.85 ^a^
Ni	0.03 ± 0.02 ^a^	0.02 ± 0.00 ^a^
Fe	2.50 ± 0.59 ^b^	8.95 ± 1.86 ^a^
B	0.12 ± 0.01 ^b^	0.16 ± 0.02 ^a^
Mn	0.39 ± 0.04 ^b^	0.46 ± 0.04 ^a^
Cr	0.02 ± 0.01 ^a^	0.01 ± 0.00 ^a^
Mo	0.02 ± 0.01 ^a^	0.02 ± 0.01 ^a^
V	0.25 ± 0.09 ^a^	0.21 ± 0.02 ^a^
Cu	0.15 ± 0.03 ^a^	0.15 ± 0.01 ^a^
Al	0.69 ± 0.43 ^a^	0.23 ± 0.08 ^b^
Sr	0.26 ± 0.15 ^a^	0.10 ± 0.01 ^b^

The mean ± standard deviation of three independent samples is shown. Different letters between the rows indicate significant differences between the mean values (Tukey) (*p* ≤ 0.05). NCCF-T: Nixtamalized commercial corn flour tortilla, NC-T: Nixtamalized corn tortilla.

**Table 5 foods-13-04082-t005:** Macromineral content in tortillas elaborated with traditional nixtamalized corn (NC-T) and nixtamalized commercial corn flour (NCCF-T) (mg/100 g tortillas).

Mineral	Macromineral Content (mg/100 g Tortilla) Samples	
	NC-T	NCCF-T
K	279.52 ± 60.37 ^b^	299.64 ± 35.23 ^a,b^
P	226.99 ± 48.34 ^a^	195.89 ± 58.76 ^a^
Ca	352.90 ± 31.28 ^a^	177.88 ± 44.99 ^a^
Mg	126.55 ± 53.33 ^a^	116.66 ± 57.73 ^a^
Na	17.72 ± 5.69 ^a^	20.22 ± 7.40 ^a^
S	16.34 ± 6.10 ^a^	9.94 ± 3.73 ^b^
Ca/P	1.55	0.90

The mean ± standard deviation of three independent samples is shown. Different letters between the columns indicate significant differences between the mean values (Tukey) (*p* ≤ 0.05). NCCF-T: Nixtamalized commercial corn flour tortilla, NC-T: Nixtamalized corn tortilla.

**Table 6 foods-13-04082-t006:** Vitamin content in NC-T and NCCF-T.

Sample	NC-T	NCCF-T
Vitamin B1 (Thiamine) (mg/kg)	1.17 ± 0.41 ^a^	2.27 ± 0.30 ^b^
Vitamin B2 (Riboflavin) (mg/kg)	0.18 ± 0.07 ^a^	0.34 ± 0.13 ^b^
Niacin (Nicotinic acid) (mg/kg)	7.24 ± 1.90 ^a^	2.28 ± 2.12 ^b^
Folic acid (Folacin) (mg/kg)	0.34 ± 0.33 ^a^	4.54 ± 0.22 ^b^

The mean ± standard deviation of three independent samples is shown. Different letters between the rows indicate significant differences between the mean values (Tukey) (*p* ≤ 0.05). NCCF-T: Nixtamalized commercial corn flour tortilla, NC-T: Nixtamalized corn tortilla.

**Table 7 foods-13-04082-t007:** Percentage of in vitro protein digestibility, total starch and resistant starch of NC-T and NCCF-T.

Sample	NC-T	NCCF-T
In vitro protein digestibility (%)	65.9 ± 3.1 ^b^	75.6 ± 1.8 ^a^
Total starch (%)	46.5 ± 11.2 ^a^	60.2 ± 13.2 ^a^
Resistant starch (%)	2.7 ± 1.0 ^a^	3.4 ± 0.9 ^a^

The mean ± standard deviation of three independent samples is shown. Different letters between the rows indicate significant differences between the mean values (Tukey) (*p* ≤ 0.05). NCCF-T: Nixtamalized corn flour tortilla, NC-T: Nixtamalized corn tortilla.

**Table 8 foods-13-04082-t008:** Contribution of the tortilla made with a mixture of enriched nixtamalized commercial corn flour (NCC-F) and traditional nixtamal masa (NM) to the recommended daily intake (RDI) of some vitamins and minerals for the Mexican population.

Micro-Nutrient	Contribution (mg/129.84 g Tortilla on Dry Basis ^Ϯ^) Mix 75:25 *	Contribution (mg/129.84 g Tortilla on Dry Basis ^Ϯ^) Mix 50:50 **	RDI (Age Group) (mg)	Contribution of the Tortilla 50:50 to the RDI (%)
Calcium	401.38	344.56	CH 800	CH 43.07
			TA 1000	TA 34.46
			A 800	A 43.07
			PW 1200	PW 28.71
			BW 1200	BW 28.71
Phosphorus	284.62	274.52	CH 500	CH 54.90
			TA 1250	TA 21.96
			A 700	A 39.21
			PW 700	PW 39.21
			BW 700	BW 39.21
Magnesium	161.10	157.89	CH 100	CH >100
			TA 400	TA 40.00
			A 350	A 45.00
			PW 320	PW 49.30
			BW 355	BW 44.47
Iron	5.34	7.44	CH 15	CH 50.00
			TA 15	TA 50.00
			A 15	A 50.00
			PW 30	PW 25.00
			BW 15	BW 50.00
Zinc	12.02	11.13	CH 15	CH 74.20
			TA 15	TA 74.20
			A 15	A 74.20
			PW 15	PW 74.20
			BW 19	BW 58.60
Vitamin B1	0.21	0.21	CH 0.8	CH 26.25
			TA 1.2	TA 17.50
			A 1.5	A 14.00
			PW 1.5	PW 14.00
			BW 1.6	BW 13.13
Vitamin B2	0.01	0.01	CH 1.0	CH <1.0
			TA 1.5	TA <1.0
			A 1.7	A <1.0
			PW 1.7	PW <1.0
			BW 1.8	BW <1.0
Vitamin B3	0.90	0.01	CH 6	CH <1.0
			TA 20	TA <1.0
			A 15	A <1.0
			PW 13	PW <1.0
			BW 15	BW <1.0
Folic acid	0.19	0.335	CH 0.15	CH >100
			TA 0.18	TA 100
			A 0.20	A >100
			PW 0.60	PW 56
			BW 0.28	BW >100

Ϯ Represents the daily per capita consumption of tortillas by the Mexican population on a wet basis (240 g/day), which corresponds to 129.84 g on a dry basis (with an average moisture content of 45.9%). CH: children, TA: adolescents, A: adults, PW: pregnant women, BW: breastfeeding woman. * Denotes a 75:25 (*w*/*w*) mixture of NM and NCC-F. ** Denotes a 50:50 (*w*/*w*) mixture of NM and NCC-F.

## Data Availability

The original contributions presented in this study are included in the article. Further inquiries can be directed to the corresponding authors.

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
