# Peer review of "Indirect Fortification of Traditional Nixtamalized Tortillas with Nixtamalized Corn Flours"

_foods, 2024, doi:10.3390/foods13244082_

Round 1
Reviewer 1 Report
Comments and Suggestions for Authors
The paper titled "Indirect fortification of traditional nixtamalized tortillas with nixtamalized corn flours" is interesting. This manuscript focused on the study of the indirect fortification of Mexican tortillas made from nixtamalized masa with nixtamalized commercial corn flour fortified with Zn, Fe, vitamins and folic acid. Title, keywords, are clearly describes what the manuscript is about. There is introduction to justify that the work is included in the scope of the journal "Foods”. Data of experiments are properly analyzed and interpreted. The authors correctly drew conclusions from the research.
Detailed comments
Abstract is not complete: should include more information regarding principal results and conclusions. The abstract is often presented separately from the article, so it must be able to stand alone.
L.61-86- information on literature sources is missing
L.99: ´’The authors reported that the corn flour tortilla….” – it is not clear which authors are reported.
Author Response
Comments and Suggestions for Authors
The paper titled "Indirect fortification of traditional nixtamalized tortillas with nixtamalized corn flours" is interesting. This manuscript focused on the study of the indirect fortification of Mexican tortillas made from nixtamalized masa with nixtamalized commercial corn flour fortified with Zn, Fe, vitamins and folic acid. Title, keywords, are clearly describes what the manuscript is about. There is introduction to justify that the work is included in the scope of the journal "Foods”. Data of experiments are properly analyzed and interpreted. The authors correctly drew conclusions from the research.
A:// Thank you for your comment. All suggested changes have been incorporated into the revised version of the manuscript.
Detailed comments
Abstract is not complete: should include more information regarding principal results and conclusions.
A:// The abstract was completed including the information requested by the reviewer.
The abstract is often presented separately from the article, so it must be able to stand alone.
A:// The abstract was modified including relevant information about the investigation work, in such a way, the lector understands the results relevance regarding to the nutritional contribution of the tortilla to the diet of the Mexican population.
L.61-86- information on literature sources is missing
A:// Thanks for the comments, adequate references were added to support the state of art presented in the introduction section.
L.99: ´’The authors reported that the corn flour tortilla….” – it is not clear which authors are reported.
A:// The text was modified to avoid the omission of references.
Reviewer 2 Report
Comments and Suggestions for Authors
The manuscript is well written. A lot of new foods using the formulation method are being introduced on the market to increase the chemical content and to meet the vitamin and mineral needs of consumers.
At the "In vitro protein digestibility" lines 192-204 I have a few questions:
A multienzymx system consisting of porcin trypsin, bovine chymotrypsin and bacterial protease was used. pH mention at line 199 and line 201 was pH=8.0 but that enzymes, similar with human digestiv tract are working at low pH near 1 not high like is described in the method. Further more, a high pH inactivated the proteolitic enzymes, except bacterial protease. Please explain why you used a high value of pH. In the same way you describe in the method that used HCl to adjusting pH value, but at pH=8 basic domain need to used NaOH not HCl for adjusting.
Line 320 you mention that temperature affect the content of lysine and tryptophan content after digestibility process, please explain the value of the temperature and how the content was influenced, the mechanism.
Comments on the Quality of English LanguageNo comments. Manuscript is well written.
Author Response
Comments and Suggestions for Authors
The manuscript is well written. A lot of new foods using the formulation method are being introduced on the market to increase the chemical content and to meet the vitamin and mineral needs of consumers.
A:// Thank you for your comments. I carefully included all the suggested changes. I hope that the modification makes the manuscript suitable for publication in Foods.
At the "In vitro protein digestibility" lines 192-204 I have a few questions:
A multienzymx system consisting of porcin trypsin, bovine chymotrypsin and bacterial protease was used. pH mention at line 199 and line 201 was pH=8.0 but that enzymes, similar with human digestiv tract are working at low pH near 1 not high like is described in the method. Further more, a high pH inactivated the proteolitic enzymes, except bacterial protease. Please explain why you used a high value of pH. In the same way you describe in the method that used HCl to adjusting pH value, but at pH=8 basic domain need to used NaOH not HCl for adjusting.
A:// Dear Reviewer, thank you for your comment. In the described experiment, pH 8.0 was likely chosen to align with the optimal activity range of the specific enzymes included in the multienzyme system. While the human digestive tract enzymes such as pepsin operate at very low pH (around 1-2 in the stomach), trypsin and chymotrypsin, which are active in the small intestine, function optimally at higher pH values, typically in the range of 7.5-8.5. The experiment might simulate intestinal rather than gastric digestion. Using pH 8.0 provides the necessary conditions to maintain the enzymatic activity of trypsin and chymotrypsin while enabling the bacterial protease to function.
You are correct that adjusting a solution to a basic pH such as 8.0 would typically require a base like NaOH rather than an acid like HCl. The mention in the methodology refers to the use of HCl and/or NaOH (0.1 N) as required to adjust the pH.
Line 320 you mention that temperature affect the content of lysine and tryptophan content after digestibility process, please explain the value of the temperature and how the content was influenced, the mechanism.
A:// Thank you for your comment. During nixtamalization and tortilla making, temperatures ranging from 80–90°C during cooking and over 200°C during baking significantly impact the lysine and tryptophan content, leading to losses of approximately 18.60% and 21.47%, respectively. These losses occur primarily due to the Maillard reaction, where lysine reacts with reducing sugars, and the alkaline hydrolysis and thermal degradation of amino acids, particularly tryptophan. The high pH environment and elevated temperatures accelerate these reactions, reducing the nutritional quality of the final product. Mitigating strategies include optimizing process conditions, such as lowering temperatures and cooking times, and fortifying tortillas to compensate for the loss of these essential amino acids.
Reviewer 3 Report
Comments and Suggestions for Authors
This manuscript discusses the indirect fortification of nixtamalized tortillas with nixtamalized corn flours fortified with Zn, Fe, vitamins, and folic acid. The topic of the work is suitable for publication in the journal. However, presentation of some of the tables is not clear to the reader as headings are missing and are not currently adequately provided. The authors should consider presenting their data clearly in the tables. Many abbreviations are also not explained in full term or accurately presented and should be provided/corrected.
I have the following comments to the authors:
1. Many abbreviations should be provided in full term or accurate presented:
· In lines 31-32, the abbreviation "NC-T" should present Nixtamalized corn tortilla. The word "corn" is missing and should be added.
· In line 34, and 102 provide the complete term for PDCAAS.
· In the footnote of Table 1, the abbreviation for NCCF-T should be provided as Nixtamalized commercial corn flour tortilla
· In the subheading 2.4, what does “OES” in “ICP-OES” stand for? Please add the complete term to inductively coupled plasma.
· In line 205, what does "IVPD" stand for? the complete term should be added.
2. In line 79, “In Mexico, it is still discussed the concept of tortilla quality”. This part of the sentence does not read well and should be rewritten.
3. Table 2 does not have a heading. Please add headings explanatory of the information presented in the two columns. what do the values in the middle column present?
4. Lines 193-197, specification of the enzymes should be added to the materials section.
5. Lines 216-217, the sentence: “The starch was starch with pancreatic -amylase (3 Ceralpha U/mg) for 16 h at neutral pH”
This sentence does not read well and should be rewritten.
6. The heading of Table 3 is not clear. Which column shows moisture and which one the protein content? In lines 263-264, the authors referred to the ash content. The values for ash content are not presented in Table 3 and the reader cannot compare the results. Please add the values to Table 3 and distinguish them with clear headings.
7. Many percentages are presented with a minus sign before the number, mainly in the abstract section. The minus signs should be removed. Please check throughout the manuscript and correct.
8. Lines 431-432, the sentence: “The mixture between fortified commercial corn flours and traditional nixtamal produce beneficial health benefices”
This sentence does not read well and should be rewritten.
Comments on the Quality of English LanguageEnglish language should be improved. A number of sentences do not read well and should be rewritten.
Author Response
Comments and Suggestions for Authors
This manuscript discusses the indirect fortification of nixtamalized tortillas with nixtamalized corn flours fortified with Zn, Fe, vitamins, and folic acid. The topic of the work is suitable for publication in the journal. However, presentation of some of the tables is not clear to the reader as headings are missing and are not currently adequately provided. The authors should consider presenting their data clearly in the tables. Many abbreviations are also not explained in full term or accurately presented and should be provided/corrected.
A:// Dear reviewer, thank you for your comments. I included all the suggested modifications. Please refer to the revised version of the manuscript.
I have the following comments to the authors:
- Many abbreviations should be provided in full term or accurate presented:
- In lines 31-32, the abbreviation "NC-T" should present Nixtamalized corn tortilla. The word "corn" is missing and should be added.
A:// It was fixed.
- In line 34, and 102 provide the complete term for PDCAAS.
A:// It was fixed.
- In the footnote of Table 1, the abbreviation for NCCF-T should be provided as Nixtamalized commercial corn flour tortilla
A:// It was fixed.
- In the subheading 2.4, what does “OES” in “ICP-OES” stand for? Please add the complete term to inductively coupled plasma.
A:// It was fixed.
- In line 205, what does "IVPD" stand for? the complete term should be added.
A:// It was fixed.
- In line 79, “In Mexico, it is still discussed the concept of tortilla quality”. This part of the sentence does not read well and should be rewritten.
A:// It was fixed.
- Table 2 does not have a heading. Please add headings explanatory of the information presented in the two columns. what do the values in the middle column present?
A:// It was fixed.
- Lines 193-197, specification of the enzymes should be added to the materials section.
A:// Dear reviewer, this information is included originally in the materials section. ‘A multienzyme system consisting of porcine pancreatic trypsin (Type IX, 15,310 units/mg protein), bovine pancreatic chymotrypsin (Type II, 48 units/g solid), porcine intestinal peptidase (P-7500, 115 units/mg solid) and bacterial protease (Type XIV, 4.4 units/mg solid) (Sigma-Aldrich Co., St. Louis, MO, USA) was used.’
- Lines 216-217, the sentence: “The starch was starch with pancreatic -amylase (3 Ceralpha U/mg) for 16 h at neutral pH”
This sentence does not read well and should be rewritten.
A:// It was fixed.
- The heading of Table 3 is not clear. Which column shows moisture and which one the protein content? In lines 263-264, the authors referred to the ash content. The values for ash content are not presented in Table 3 and the reader cannot compare the results. Please add the values to Table 3 and distinguish them with clear headings.
A:// Dear Reviewer, Table 3 shows the values of tortillas made with traditional nixtamalized corn (NC-T) and nixtamalized corn flour (NCCF-T). The ash content is originally indicated in the table and the significant difference is indicated by the different letters in the rows. Please refer to table 3.
- Many percentages are presented with a minus sign before the number, mainly in the abstract section. The minus signs should be removed. Please check throughout the manuscript and correct.
A:// Dear reviewer, the minus sign ‘-’ is the symbol ‘~’, which stands for 'approximately', 'around' or 'about'.
- Lines 431-432, the sentence: “The mixture between fortified commercial corn flours and traditional nixtamal produce beneficial health benefices”
This sentence does not read well and should be rewritten.
A:// It was corrected.
Comments on the Quality of English Language
English language should be improved. A number of sentences do not read well and should be rewritten.
A:// Thank you for your comments. The English grammar and editing of the manuscript have been carefully revised.
Round 2
Reviewer 3 Report
Comments and Suggestions for Authors
My comments to the authors are mostly addresses. However still sentences do exist in the manuscript that are wrongly structured and cannot convey the meaning as presented. These sentences are required to be rewritten. Errors still exist in the work that should be corrected.
The authors should consider correcting the following points:
1. In line 39 of the abstract, please check if the value of 50% RDI for folic acid is correctly presented based on the information provided in Table 8.
2. Referring to lines 42-44, the sentence: “Tortillas made from a mixture of NCC-F and traditional NM are an effective strategy to combat micronutrient deficiencies in the Mexican population
The sentence cannot convey the meaning as presented. Are tortillas considered as a strategy? This sentence should be rewritten.
3. Referring to lines 80-82, the sentence: “This fact is even confusing for the consumer, due to in tortillas made with 100% of corn, in most cases are not packaged; therefore, no possibility to difference between the quality of the masa and the flour”
This sentence is grammatically inaccurate, cannot be understood, and should be rewritten.
4. In the heading of Table 2, what do authors mean by the word "nutrimental"? Do they intend to refer to nutritional?
5. In subheading 2.3, a typing error exists and “optimcal” should be corrected to “optical”
6. In line 360, the parenthesis after RDI should be removed.
Comments on the Quality of English LanguageErrors exist in sentence structure. Still a number of sentences are grammatically inaccurate, wrongly structured and should be rewritten (please see my comments).
Author Response
My comments to the authors are mostly addresses. However still sentences do exist in the manuscript that are wrongly structured and cannot convey the meaning as presented. These sentences are required to be rewritten. Errors still exist in the work that should be corrected.
A:// Dear reviewer, thank you for your comments. All suggested changes have been incorporated into the revised version of the manuscript.
The authors should consider correcting the following points:
- In line 39 of the abstract, please check if the value of 50% RDI for folic acid is correctly presented based on the information provided in Table 8.
A:// It was fixed
- Referring to lines 42-44, the sentence: “Tortillas made from a mixture of NCC-F and traditional NM are an effective strategy to combat micronutrient deficiencies in the Mexican population
The sentence cannot convey the meaning as presented. Are tortillas considered as a strategy? This sentence should be rewritten.
A:// Thank you for your comment. It was rewritten.
- Referring to lines 80-82, the sentence: “This fact is even confusing for the consumer, due to in tortillas made with 100% of corn, in most cases are not packaged; therefore, no possibility to difference between the quality of the masa and the flour”
This sentence is grammatically inaccurate, cannot be understood, and should be rewritten.
A:// Thank you for your comment. It was rewritten.
- In the heading of Table 2, what do authors mean by the word "nutrimental"? Do they intend to refer to nutritional?
A:// Thank you for your comment. It was a typographical error that has been corrected.
- In subheading 2.3, a typing error exists and “optimcal” should be corrected to “optical”
A:// It was fixed.
- In line 360, the parenthesis after RDI should be removed.
A:// It was fixed.
Thank you for carefully reviewing the manuscript and helping to improve its quality.